# An In Silico Feasibility Study of Dose-Escalated Hypofractionated Proton Therapy for Rectal Cancer

**DOI:** 10.3390/cancers17162627

**Published:** 2025-08-11

**Authors:** Erik Almhagen, Ali Alkhiat, Bruno Sorcini, Freja Alpsten, Camilla J. S. Kronborg, Heidi S. Rønde, Marianne G. Guren, Sara Pilskog, Alexander Valdman

**Affiliations:** 1Department of Nuclear Medicine and Medical Physics, Karolinska University Hospital, 14186 Stockholm, Sweden; 2Department of Oncology-Pathology, Karolinska Institute, 17177 Stockholm, Sweden; 3Danish Centre for Particle Therapy, Aarhus University Hospital, 8200 Aarhus, Denmark; 4Department of Clinical Medicine, Aarhus University, 8200 Aarhus, Denmark; 5Department of Oncology, Oslo University Hospital, 0424 Oslo, Norway; 6Institute of Clinical Medicine, University of Oslo, 0424 Oslo, Norway; 7Cancer Clinic, Haukeland University Hospital, 5009 Bergen, Norway; 8Institute of Physics and Technology, University of Bergen, 5007 Bergen, Norway; 9Department of Radiation Oncology, Karolinska University Hospital, 17176 Stockholm, Sweden

**Keywords:** proton therapy, hypofractionation, rectal cancer

## Abstract

The incidence of colorectal cancer is increasing in young adults. Treatment often includes extensive surgery with burdensome late side effects. Radiation therapy along with chemotherapy is often used in combination before surgery and, in some cases, renders surgery unnecessary. The purpose of this study is to investigate the possibility of delivering larger radiation doses to patients without elevating the dose to healthy organs and tissue in the vicinity of the rectum. We considered two modalities of radiation: protons and photons. Our results indicate that it should be safe to give these higher doses with both radiation modalities. With protons the dose given to healthy tissue can be approximately reduced by half, compared to photons. A clinical trial would be required to verify the efficacy of larger doses to the tumor.

## 1. Introduction

Colorectal cancer is the third most common form of cancer globally, with increasing rates among young adults [1,2,3]. The standard of care for locally advanced rectal cancer (LARC) is neoadjuvant chemoradiotherapy (CRT) or total neoadjuvant therapy (TNT) followed by total mesorectal excision (TME) [4,5,6,7]. TME may come with significant sequelae [8], with recent advances in surgical techniques showing no significant improvement in post-surgery quality of life [9].

The radiotherapy (RT) doses typically given in long-course radiotherapy (LCRT) are 45–50 Gy in 1.8–2 Gy fractions [10,11]. Increasing the RT dose may increase clinical complete response (cCR) rates, with one study suggesting elevated tumor dose response in the 50.4–70 Gy interval [12]. In addition, increasing the dose may be particularly important in unresectable cases [13]. If cCR is achieved, it may be followed by a watch and wait approach [14,15] or a local excision with reduced sequelae compared to TME [16,17,18]. Although increasing RT doses may lead to increased toxicity, a meta-analysis found no increase in grade 3 toxicities with dose-escalated RT as compared to conventionally fractionated RT [19].

Proton therapy (PT) is a modality that may significantly reduce the integral dose (ID) compared to conventional photon radiotherapy (xRT) [20]. This is particularly important for younger patients with longer expected remnant lifespans, where a lower ID may reduce the risk of radiation-induced cancers [21,22]. Planning studies have identified reduced doses to the bladder, femoral heads, and bowel bag as possibly significant advantages of PT over xRT [23,24,25] for rectal cancer. These advantages may enable a more aggressive chemotherapy treatment [26]. Reduced organ at risk (OAR) doses may allow for even further dose escalation with PT, and thus increased cCR, with tolerable toxicity levels [27]. Clinical trials confirming these findings are lacking; however, one ongoing study investigates dose-escalated proton reirradiation for rectal cancer (RE-RAD II).

The increased costs and more limited access to PT as compared to xRT may make the former unfeasible, especially with LCRT; hypofractionated alternatives may be more cost-effective [28]. A short course radiotherapy (SCRT) approach, giving 25 Gy in 5 fractions, has been found to be a viable alternative to LCRT, both alone and as part of TNT, in terms of tumor control and OAR toxicity [29,30]. The optimal fractionation scheme is yet to be determined. However, SCRT is the preferred RT approach as part of the total neoadjuvant treatment in Sweden. A moderately low *α*/*β* ratio of 5 [31] indicates improved tumor control by using hypofractionation, but the same study found increased loco-regional control rates for hyperfractionation. A current phase I study is ongoing [32], comparing 5 × 6 Gy, 5 × 7 Gy, and 5 × 8 Gy schemes using xRT.

The ongoing PRORECT (NCT04525989) phase II trial [33] compares PT and xRT SCRT. The purpose of this article is to assess the feasibility of dose-escalated PT SCRT compared to xRT SCRT in preparation for a Nordic follow up study, PRORECT II. In the PRORECT II study, the primary tumor will be prescribed a dose of 5 × 6 Gy, and pathological lateral lymph nodes (LLN) will be prescribed a dose of 5 × 7 Gy in a simultaneous integrated boost (SIB), with the remaining part of the CTV receiving 5 × 5 Gy. In this we are following the approach of the NORMAL-R study [34,35]. The feasibility will be assessed in terms of achievable OAR doses and integral dose, given that target coverage is compliant with a set of criteria based on local practice at Karolinska University Hospital and the PRORECT study protocol.

## 2. Materials and Methods

### 2.1. Patient Data

Ten patients, already treated as part of the PRORECT study, were included in this study. The original planning computed tomography (CT) and structure sets were used, which include the CTV_25Gy_ structure. The structures were delineated in accordance with the PRORECT study, described previously [33] and in the PRORECT study protocol radiotherapy appendix [36]. All planning CTs were taken in the head-first supine position; slice thickness was 3 mm, with pixel sizes of 1 × 1 mm^2^. To facilitate dose planning, a physician delineated a gross tumor volume (GTV) for pathological lateral lymph nodes, around which a CTV_35Gy_ structure was delineated using a 10 mm margin. Around the primary tumor GTV, a CTV_30Gy_ structure was defined using a 15 mm margin. Both CTV_30Gy_ and CTV_35Gy_ are fully enclosed by CTV_25Gy_. Details on the patient cohort are found in Table 1.

### 2.2. Treatment Planning

All treatment planning was performed in the Eclipse (Varian Medical Systems, Palo Alto, CA, USA) treatment planning system (TPS), v16.1. Two plans were created for each treatment modality: a standard PRORECT plan and a dose-escalated PRORECT II plan. For PRORECT plans, the CTV_25Gy_, including the primary tumor and all lymph nodes, was prescribed a dose of 5 × 5 Gy RBE. For PRORECT II plans, in addition to the 5 × 5 Gy RBE prescription to the CTV_25Gy_, the CTV_30Gy_ was prescribed a dose of 5 × 6 Gy RBE, and the CTV_35Gy_ was prescribed a dose of 5 × 7 Gy RBE.

For xRT, volumetric modulated arc therapy (VMAT) plans were created using two arcs with a single isocenter. Proton plans were created using two posterior oblique fields to keep the integral dose low whilst allowing for some rectal distension [37], optimized with a single field uniform dose (SFUD) approach for all cases except one, in which multifield optimization (MFO) was used. Gaseous areas in the vicinity of the target volumes were delineated, and their Hounsfield unit (HU) values were overridden and set to water equivalent.

For the VMAT plans, planning target volumes (PTV) with an isotropic CTV-to-PTV margin of 6 mm were created. Proton plans were robustly optimized using a robustness margin of 6 mm by displacing the isocenter parallel and anti-parallel to the cardinal axes and a HU value uncertainty of ±3.5% by increasing and decreasing the HU values of the planning CT accordingly. This results in a set of 14 robustness scenarios (12 with combined isocenter displacements and HU value changes, and 2 with HU value changes exclusively). Figure 1 illustrates dose distributions of an xRT and a PT PRORECT II plan.

### 2.3. RBE and Integral Dose

Relative biological effectiveness (RBE) was assumed to be a constant 1.1 for all proton plans during treatment planning, in line with AAPM recommendations [38]. All proton doses referred to in this article are RBE-weighted by a constant factor of 1.1.

There is some evidence of RBE decreasing with increasing fraction doses [38,39,40]. To investigate whether this may lead to underdosage due to RBE decreasing below 1.1, RBE distributions were calculated. Since tools for linear energy transfer (LET) and RBE calculations are not available in our version of Eclipse, we exported all treatment plans to a research version of RayStation, v14.0.100 (Raysearch Laboratories, Stockholm, Sweden). Furthermore, RayStation was used to compute a dose-weighted LET distribution. RBE was calculated using(1)$$\mathrm{RBE}=\frac{-\alpha /\beta +\sqrt{{\left(\alpha /\beta \right)}^{2}+4D\cdot {\mathrm{RBE}}_{\max}\cdot \alpha /\beta +4D^{2}\cdot {\mathrm{RBE}}_{\min}^{2}}}{2D}$$
where D is the proton physical fraction dose, *α*/*β* is the tissue-dependent LQ-parameter ratio, and ${\mathrm{RBE}}_{\max}$ and ${\mathrm{RBE}}_{\min}$ are LET and tissue-dependent parameters calculated by using the McNamara model [41]. We used *α*/*β* = 5 [31] for the GTV, while for all other tissue *α*/*β* = 3. No LET optimization of the treatment plans was performed.

As a metric for integral dose (ID), we used(2)$$\mathrm{ID}=\sum_{i}{{RBE}_{i}\cdot D}_{i}\cdot V_{i}$$
where $D_{i}$ is the dose to the *i*:th voxel, ${RBE}_{i}$ is the RBE of the *i*:th voxel, and $V_{i}$ is the volume of the *i*:th voxel. For xRT plans, RBE = 1.0, whereas for PT plans, RBE = 1.1.

### 2.4. Plan and Feasibility Evaluation

To compare plans with differing fractionation, EQD2 calculations were carried out, with *α/β* = 5 for the GTV and *α/β* = 3 for all other tissues. All EQD2-dose evaluation was carried out in RayStation. To evaluate the feasibility of the proposed PRORECT II study dose prescriptions, target coverage was ensured to satisfy the local Karolinska criteria, supplemented by PRORECT study protocol criteria. The appropriateness of these follows from preliminary results of the ongoing PRORECT trial, indicating the absence of any inferior clinical effects of PT compared to xRT (unpublished data). For the PT plans, robust evaluation of the CTV was used with the exact same scenarios as during robust optimization. With plans satisfying these criteria, OAR doses were compared to objectives in the PRORECT study protocol radiotherapy appendix, supplemented by OAR objectives used locally at the Karolinska University Hospital. Target coverage criteria and OAR objectives are found in Table 2. None of the OAR objectives are to be interpreted as hard constraints; target coverage takes precedence. To assess the statistical significance of the differences between xRT and PT plans in the PRORECT and PRORECT II plans, Wilcoxon signed-rank tests were carried out.

## 3. Results

Dose volume coverage maps for PT and xRT plans are shown in Figure 2 and Figure 3, respectively. These show the fraction of the number of DVH curves above a certain point. For the PRORECT PT plans in patient 9, two robustness scenarios had *D*_98%_ = 94.6% for CTV_25Gy_. All other robustness scenarios for all patients, CTV volumes, and both PRORECT and PRORECT II PT plans satisfied *D*_98%_ > 95%. All robustness scenarios satisfied *D*_2%_ < 105% for the PRORECT PT plans and CTV_25Gy_. For CTV_25Gy_ in the PRORECT II PT plans, the enclosing of CTV_30Gy_ and CTV_35Gy_ led to significantly higher doses to parts of the target, as visible in Figure 2, such that the *D*_2%_ < 105% failed for all PRORECT II PT plans. This is not the case for CTV_30Gy_ and CTV_35Gy_, which do not overlap. For patients 5 and 7 the close proximity of CTV_30Gy_ and CTV_35Gy_ led to the failure of the *D*_2%_ < 105% objective for 9 and 4 robustness scenarios for patients 5 and 7, respectively, for CTV_30Gy_.

All PTV criteria in Table 1 for all xRT plans are satisfied, with the exception of the $D_{2\%}$ criteria for PTV_25Gy_ in the PRORECT II xRT plans, which failed for the same reasons as for the PT cases.

Distributions of OAR EQD2 doses for all ten patients are visualized in a set of boxplots in Figure 4. In general, PT achieves lower mean doses compared to xRT. For two patients, the PRORECT II plans for both modalities exceeded the bowel bag $D_{{5\mathrm{cm}}^{3}}$ objective. The PT median bowel bag $D_{{5\mathrm{cm}}^{3}}$ dose is higher than for xRT in the PRORECT II plans, although the difference is not statistically significant (see Table 3) at the 5% level. Exceeding this dose may lead to grade 3+ enteritis/obstruction [42]. In general, results are more similar between xRT and PT plans for the doses to small volumes/maximum doses. This is due to the similar or slightly worse lateral penumbra for protons compared to photons.

For one patient, the PRORECT II xRT plan exceeded the bladder mean dose objective. This objective is relatively strict; data from the PACE B study showed no statistically significant association with D_40%_ < 18.1 Gy in five fractions and acute/late genitourinary toxicity [43]. We would not expect a large increase in toxicity due to the meager exceedance of the dose objective for the one PRORECT II plan. The volumetric DVH criteria for the bowel bag, with dose distributions shown in Figure 5, are not exceeded in any plan. The PT plans have slightly higher median bowel bag $V_{40\mathrm{Gy}}$ doses, while the xRT plans have higher median $V_{23.8\mathrm{Gy}}$ doses.

Table 3 shows the Wilcoxon signed-rank test *p*-values. These result from a paired data test, where for each OAR dose objective the ten xRT and ten PT doses were compared, once for the PRORECT plans and once for the PRORECT II plans. Lower *p*-values were seen for the mean doses where the ID-reducing properties of PT yield larger differences. For the maximum/near maximum doses, the *p*-values are generally higher. This is in line with results seen in Figure 4 and Figure 5.

Figure 6 shows the distribution of RBE values for the union of CTV_30Gy_ and CTV_35Gy_ for each of the ten patients. The range for both PRORECT and PRORECT II plans, rounded to two decimals, is [1.09–1.16]. For the PRORECT plans, 78.7% of the number of voxels has RBE ≥ 1.1; for the PRORECT II plans, this number is 76.3%. Utilizing a Wilcoxon signed-rank test of the voxel pairs of the PRORECT and PRORECT II plans yields a *p*-value < 0.0001.

The ID for each patient, modality, and plan type is shown in Figure 7. The mean xRT/PT-ID ratio for both PRORECT and PRORECT II plans is 1.97, with a standard deviation of 0.26. The Wilcoxon signed-rank test resulted in *p*-values of 0.002 when comparing both PRORECT and PRORECT II plans, respectively, indicative of a statistically significant difference. For patient 1, the PRORECT II PT plan was MFO-optimized, leading to a slightly lower ID than the SFUD-optimized PRORECT PT plan. Due to the PT potential for modulation along the beam central axis, the IDs for PT plans are sensitive to increases in CTV size both in the superior–inferior direction as well as the anterior–posterior direction; for xRT plans this sensitivity is greater in the superior–inferior direction than the anterior–posterior direction.

## 4. Discussion

It is our clinical experience that it is relatively easy to create PRORECT treatment plans that achieve all target coverage and OAR objectives. This is suggestive of the potential for dose escalation. For the PRORECT II plans, although sufficient levels of target coverage were achieved, in this case some OAR objective doses were exceeded. We believe this indicates the appropriateness of the magnitude of the PRORECT II dose escalation.

One study found that for PT, two posterior oblique fields is the most robust against rectal distension [37], which is what we used in our PT plans. We employed robust optimization and an SFUD planning approach in the majority of our PT plans to increase robustness [44]. Furthermore, by overriding any gaseous areas and forcing the optimizer to consider those areas as water equivalent, we mitigate the risk of undershooting, i.e., protons stopping short of the distal target edge, but increase the risk of overshooting, which may increase the dose to distal OARs such as the bladder and bowel bag.

There are no PT dose escalation clinical trials for primary rectal cancer known to us at this time. Although this planning study and others [23] indicate the favorable dosimetric properties of PT over xRT for rectal cancer in some cases, PT-specific uncertainties may lead to in-patient dose distribution degradation. Due to bowel motion and rectal distention and the breakdown of the static dose cloud approximation for PT, dosimetric uncertainties are present with PT that planning studies such as the present one may fail to account for. In an LCRT setting the averaging effect over many fractions may alleviate the effect of these uncertainties. However, in an SCRT setting as considered here, the magnitude of the PT uncertainties may be larger. The small size of the CTV_35Gy_ structure and the small number of fractions may in unfortunate circumstances lead to lower cumulative doses than those prescribed. A dose accumulation study based on daily imaging may shed light on this issue.

Although there is some evidence for a dose escalation [12], the RECTAL-BOOST phase II trial found no evidence of increased cCR/pCR rates with dose escalation up to 65 Gy [45]. However, the trial did find an increase in near-complete response rates. It differs from our approach here since an LCRT approach was used with 25 × 2 Gy fractions up to 50 Gy and then 5 × 3 Gy fractions up to 65 Gy to the GTV; furthermore, target coverage was compromised to ensure OAR dose criteria were fulfilled. In our feasibility study, we generally only boost a small volume up to a dose of 7 × 5 Gy RBE, leading to a dose of 60 Gy EQD2 (*α*/*β* = 5). OAR dose objectives were exceeded in only a few cases, and even when they were, target coverage criteria were not compromised.

The reduction in ID for PT may be significant for younger patients with a longer life expectancy, as there is evidence of a reduced risk of secondary cancer induction when using PT as compared to xRT [22,46]. The clinical effects of the lower mean doses for PT compared to xRT remain unclear, but the limiting of a dose bath effect may be beneficial for reirradiation [47,48,49,50]; one study found a local recurrence rate for LARC of 5% [51]. Although not used in this feasibility study, LET optimization in PT could potentially increase the biological efficiency of PT in the target compared to xRT, even if posterior oblique fields are used exclusively [52]. With LET optimization, MFO would be necessary, although the previously cited study indicated acceptable levels of robustness even with MFO.

Per Equation (1), RBE is a function of fraction dose, LET, and *α*/*β*. In following the AAPM recommendation to assume RBE = 1.1 [38] during treatment planning, there is the possibility of delivering a lower biologically weighted dose to the target due to the high fraction doses. In general, the PRORECT plans yield higher RBE values in the escalated CTV voxels compared to PRORECT II plans. This is in line with in vitro data [40], as well as with Equation (1) for which $\underset{D\to \infty }{\lim}\mathrm{RBE}={\mathrm{RBE}}_{\min}$ [53]. However, the McNamara model suggests that deviations from RBE = 1.1 in the escalated CTV voxels are relatively small, as are the RBE differences between the PRORECT and PRORECT II plans. With preliminary results from the PRORECT trial not indicating inferiority of PT compared to xRT, to the extent RBE = 1.1 is appropriate for the 5 Gy fractions of PRORECT, the McNamara model would suggest it is appropriate also for the 6 and 7 Gy fractions of PRORECT II. However, RBE models in PT are associated with large uncertainties, and the inter-model variation in predicted RBE is large [54].

The establishing of clinical benefits of PT with its lower ID and mean doses compared to xRT, especially in light of the previously mentioned uncertainties, necessitates a clinical trial, which is the object of the upcoming PRORECT II study. In an SCRT approach, dose escalation will invariably lead to relatively large fraction doses. An interesting and important secondary endpoint for a clinical trial would be any differences in tumor control relative to xRT due to in vivo RBE deviating significantly from 1.1.

A limitation of this study is the size of our patient cohort. Using ten patients may limit the generalizability of our results. However, we believe ten patients is sufficient for our purposes with this article. As shown in Table 1, there is substantial variation in tumor heights, volumes, and staging in the cohort. Furthermore, the clinical NORMAL-R study [35], which we used as a basis for our proposed dose escalation, only used twenty patients in its cohort. Furthermore, since this is an in silico feasibility study ahead of the actual clinical trial, no clinical outcome data can be provided to assess toxicity levels that would result from the PRORECT II plans.

## 5. Conclusions

The achievable target coverage for the PRORECT II treatment plans, in combination with the acceptable doses to the OAR, shows the feasibility of the proposed PRORECT II dose escalation. PT can reduce the ID by a factor of two compared to xRT, which may be especially important for the growing numbers of young patients to reduce the risk of late radiation-induced malignancies. RBE calculations suggest that deviations from RBE = 1.1 inside the target are small, both for the PRORECT and PRORECT II plans. A clinical trial, PRORECT II, would be necessary to establish conclusively the standard RT modality and fractionation for treatment of LARC.

## Figures and Tables

**Figure 1 cancers-17-02627-f001:**
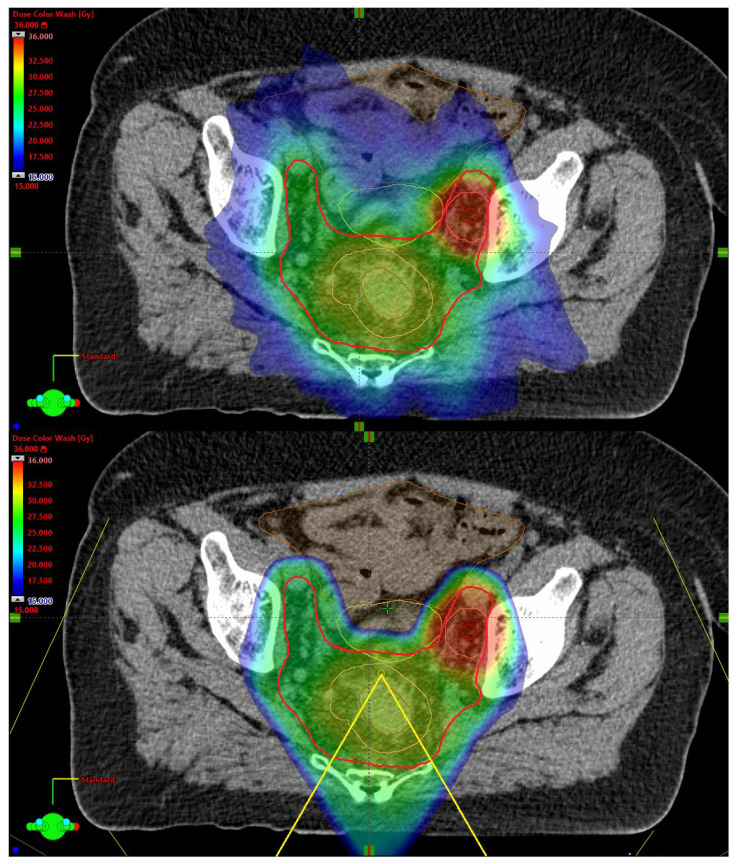
Examples of in-patient dose distributions for PRORECT II plans. The large bright red structure is the CTV_25Gy_. The central pink structure is the CTV_30Gy_, encompassing the primary tumor GTV. The pink structure on the right is the CTV_35Gy_, encompassing the GTV of the pathological lateral lymph node. The dose distribution shown ranges from 15 Gy in dark blue to 36 Gy in dark red. (**upper**) The xRT dose distribution. (**lower**) The PT dose distribution, with the field angles shown in yellow.

**Figure 2 cancers-17-02627-f002:**
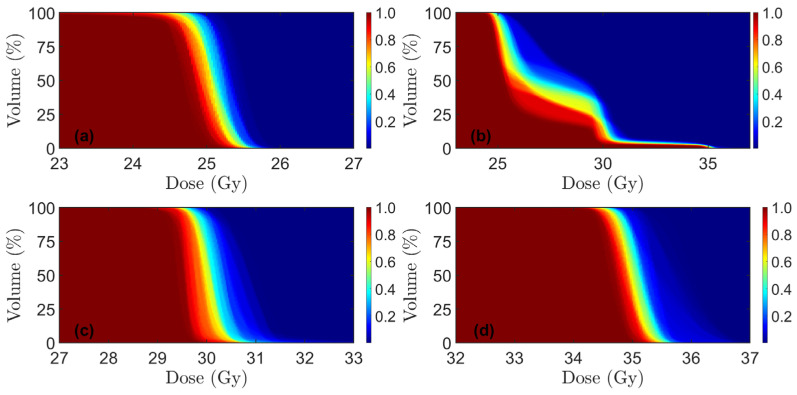
Dose volume coverage maps for PT, showing the fraction of robustness scenarios pooled for all ten patients (N = 140) in which the DVH curve lies above a certain point. Note that x-axis scales differ among the plots. (**a**) PRORECT plan, CTV_25Gy_; (**b**) PRORECT II plan, CTV_25Gy_ (**c**) PRORECT II plan, CTV_30Gy_; (**d**) PRORECT II plan, CTV_35Gy_.

**Figure 3 cancers-17-02627-f003:**
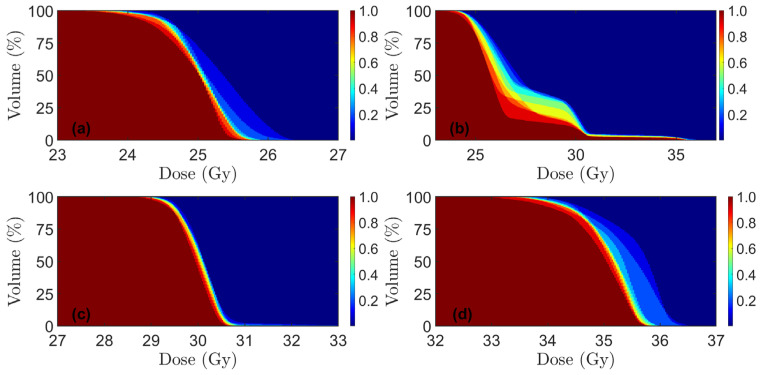
Dose volume coverage maps for xRT, showing the fraction of treatment plans pooled for all ten patients (N = 10), in which the DVH curve lies above a certain point in dose-volume space. Note that x-axis scales differ among the plots. (**a**) PRORECT plan, PTV_25Gy_; (**b**) PRORECT II plan, PTV_25Gy_; (**c**) PRORECT II plan, PTV_30Gy_; (**d**) PRORECT II plan, PTV_35Gy_.

**Figure 4 cancers-17-02627-f004:**
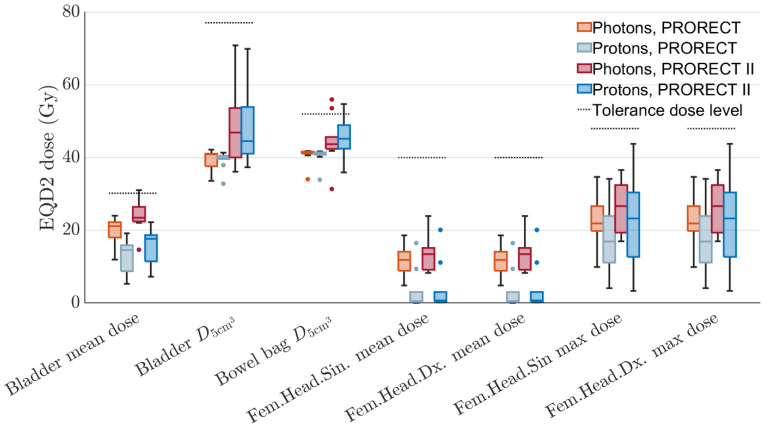
Distribution of EQD2 doses for all ten patients, for different OARs. Dotted horizontal lines show the tolerance EQD2 dose for each OAR and DVH metric.

**Figure 5 cancers-17-02627-f005:**
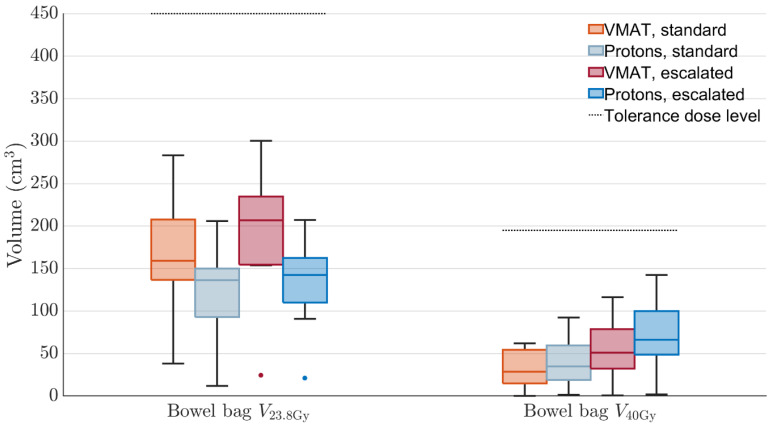
Distribution of volumes receiving DVH criteria EQD2 doses for all ten patients, for the bowel bag. Dotted horizontal lines show the tolerance volume.

**Figure 6 cancers-17-02627-f006:**
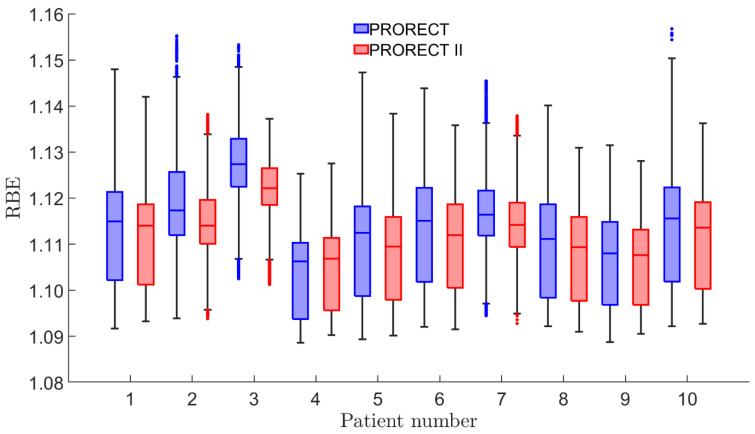
Boxplots of the distribution of RBE values, in the voxels of the union of escalated CTVs (i.e., CTV_30Gy_ and CTV_35Gy_), for each patient.

**Figure 7 cancers-17-02627-f007:**
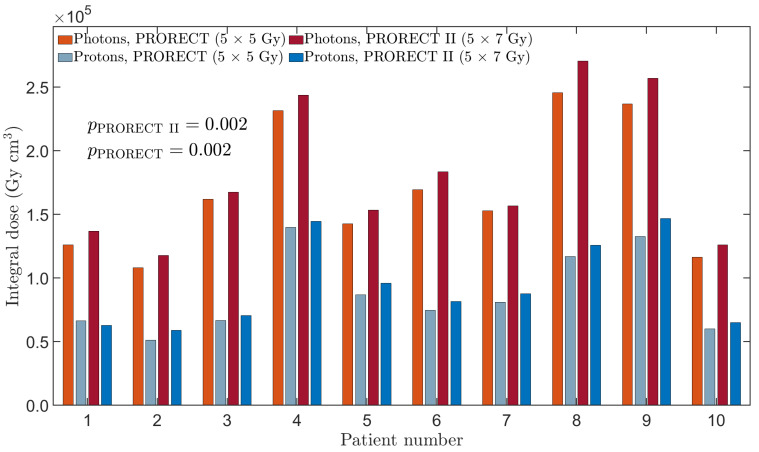
The integral dose for each patient, plan type and modality, calculated using Equation (2).

**Table 1 cancers-17-02627-t001:** Baseline clinical characteristics of the patients in the study.

Patient	Sex	Age	Tumor Height, mm	Lateral LN+	mr T-Stage	mr N-Stage	MRF+	EMVI+	LN + Compartment *	Inguinal Lymphatic Nodes in CTV	Iliaca Externa Subsite in CTV
1	Male	47	90	0	T4a	N2	1	0	1	0	0
2	Female	51	85	0	T3b	N1	0	1	2	0	0
3	Male	62	50	1	T2	N1	0	0	2	0	0
4	Male	48	30	0	T4b	N2	1	1	1	1	0
5	Female	73	60	0	T4b	N2	1	1	2	0	1
6	Male	61	100	0	T3c	N2	1	1	2	0	0
7	Male	51	120	0	T3b	N1	0	1	1	0	0
8	Male	60	70	1	T3a	N2	0	0	1	0	0
9	Female	58	20	0	T4b	N1	1	1	1	1	1
10	Male	68	60	0	T3c	N2	0	0	2	0	0

* 1 = Iliaca interna, 2 = obturator.

**Table 2 cancers-17-02627-t002:** DVH criteria for all plans. All absolute doses are given in EQD2. Target coverage criteria take precedence over OAR objective criteria.

Modality	Structure	DVH Criteria
PT	All CTV volumes	$D_{90\%}>95\%$ for all robustness scenarios
PT	All CTV volumes	$D_{98\%}>95\%$ for 12/14 robustness scenarios
PT	All CTV volumes	$D_{2\%}<105\%$ for 12/14 robustness scenarios
xRT	All PTVs	$D_{98\%}>95\%$
xRT	All PTVs	$D_{2\%}<105\%$
Both	Bowel bag	$V_{23.8\mathrm{Gy}}<450\;{\mathrm{cm}}^{3}$
Both	Bowel bag	$V_{40\mathrm{Gy}}<195\;{\mathrm{cm}}^{3}$
Both	Bowel bag	$D_{{5\mathrm{cm}}^{3}}<52\;\mathrm{Gy}$
Both	Bladder	$D_{\mathrm{mean}}<30.2\;\mathrm{Gy}$
Both	Bladder	$D_{{5\mathrm{cm}}^{3}}<77.1\;\mathrm{Gy}$
Both	Femoral head	$D_{\mathrm{mean}}<40\;\mathrm{Gy}$
Both	Femoral head	$D_{\max}<48\;\mathrm{Gy}$

**Table 3 cancers-17-02627-t003:** *p*-values resulting from comparing proton and photon plans in the PRORECT and PRORECT II arms, respectively. *p*-values that are below 5% significance levels are in bold.

Metric	PRORECT (xRT vs. PT)	PRORECT II (xRT vs. PT)
Bladder mean dose	**0.002**	**0.004**
$\mathrm{Bladder}\;D_{{5\mathrm{cm}}^{3}}$	0.492	0.625
$\mathrm{Bowel}\;\mathrm{bag}\;D_{{5\mathrm{cm}}^{3}}$	0.193	0.375
Fem. head left, mean dose	**0.002**	**0.002**
Fem. head right, mean dose	**0.002**	**0.002**
Fem. head left, max dose	**0.037**	0.106
Fem. head right, max dose	**0.037**	0.106
$\mathrm{Bowel}\;\mathrm{bag}\;V_{23.8\mathrm{Gy}}$	**0.002**	**0.002**
$\mathrm{Bowel}\;\mathrm{bag}\;V_{40\mathrm{Gy}}$	0.322	**0.014**

## Data Availability

The patient data cannot be made available due to privacy and ethical concerns. Post-processed data, such as DVHs, based on the patient data are available from the corresponding author upon reasonable request.

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
