# Peer review of "An In Silico Feasibility Study of Dose-Escalated Hypofractionated Proton Therapy for Rectal Cancer"

_cancers, 2025, doi:10.3390/cancers17162627_

Round 1

Reviewer 1 Report

Comments and Suggestions for Authors

This is an innovative and interesting feasibility study on proton therapy in rectal cancer. Proton beam therapy has emerged as an alternative to conventional radiotherapy, aiming to reduce the risk of radiation exposure to surrounding organs. However, the data on oncological outcomes associated with the use of proton beam therapy in rectal cancer remain scarce. The authors emphasize that the dose of radiotherapy given to healthy tissues may be reduced by half, which is important, especially in young adults. They included 10 patients with locally advanced rectal cancer that previously received 5x5 Gy radiotherapy. Two proton plans and two photon plans were created for each patient. The methodology is described meticulously.  All tables and figures are informative and clearly present the findings. 

There are only minor issues that should be addressed before publication:

1) The study included patients after 5x5Gy radiotherapy - some of them received proton therapy, which should be included in Table 1. What was the rationale behind the treatment planning after radiotherapy? Maybe the results would be more reliable when the planning of dose-escalation treatment were performed in patients with LARC but without previous radiotherapy?

2) In lines 108, 190, and 242, citations are missing. 

To sum up, the study provides solid justification for PRORECT II trial. 

Author Response

Comments 1: The study included patients after 5x5Gy radiotherapy - some of them received proton therapy, which should be included in Table 1. What was the rationale behind the treatment planning after radiotherapy? Maybe the results would be more reliable when the planning of dose-escalation treatment were performed in patients with LARC but without previous radiotherapy?

Response 1: We thank the reviewer for this comment. All 10 patients in this cohort received RT in the PRORECT trial. This is a retrospective in silico treatment planning feasibility study. All data, i.e. CT images and structure sets, used for the treatment planning in this study were obtained prior to treatment. The results of our study should therefore not be influenced by the fact that the patients subsequently received RT. We therefore feel no change to the manuscript is warranted here. 

Comments 2: In lines 108, 190, and 242, citations are missing

Response 2: Citations corrected in revised manuscript by Cancers editor.

Reviewer 2 Report

Comments and Suggestions for Authors

The term 'feasibility study' refers to a clinical study.

I would have suggested that the term 'in silico feasibility study' is more appropriate.

I have no further comments

Author Response

Comments 1: The term 'feasibility study' refers to a clinical study.

I would have suggested that the term 'in silico feasibility study' is more appropriate.

Response 1: We thank the reviewer for this comment, and we agree with this. Manuscript title updated to include the term "in Silico".

Reviewer 3 Report

Comments and Suggestions for Authors

This planning study investigated the feasibility of dose-escalated proton therapy (PT) for locally advanced rectal cancer (LARC). While the methodology is generally sound, several issues require clarification and improvement.

  1. As noted in the Introduction (Line 82), the optimal fractionation for LARC remains undetermined, yet long-course radiotherapy (LCRT) dominates clinical practice. Please elaborate why this study exclusively focuses on short-course radiotherapy (SCRT), particularly given SCRT's debated efficacy in dose-escalation contexts versus LCRT.
  2. It is suggested to explicitly state the purpose of this study in last paragraph of Introduction.
  3. Given substantial heterogeneity in LARC (e.g., tumor location, cT/N stage, tumor volume), the sample size (n = 10) of this study raises concerns about generalizability. Could the sample be expanded?
  4. Results are currently primarily observational. It would be advisable to incorporate statistical tests to compare the differences among the groups and determine whether they are statistically significant, thereby enhancing the persuasiveness of the findings.
  5. It is suggested that a section discussing the limitations of this study be added at the end of Discussion.
  6. In Figure 3, for bowel bag D5cm³, the Protons, PRORECT II group exhibits higher EQD2 doses than other plans, with its upper quartile exceeding tolerance. Please explain this finding and its potential implication.
  7. Figure 4 appears truncated in the provided PDF. Ensure all labels/legends are fully visible.
  8. Please check for some typographical errors, such as those on lines 108-109 and line 190, as well as the Figure 7 in line 214.

Author Response

Comments 1: As noted in the Introduction (Line 82), the optimal fractionation for LARC remains undetermined, yet long-course radiotherapy (LCRT) dominates clinical practice. Please elaborate why this study exclusively focuses on short-course radiotherapy (SCRT), particularly given SCRT's debated efficacy in dose-escalation contexts versus LCRT.

Response 1: Both SCRT and long-course chemoradiation therapy (LCRT) are the two established preoperative radiotherapeutic modalities that are equally effective in lowering risk of developing local recurrencies. Both ESMO and NCCN guidelines list either SCRT or LCRT followed by systemic chemotherapy as options for preoperative treatment in LARC.

SCRT for rectal cancer was pioneered in Sweden and has since gained increased acceptance in treatment of LARC. Currently, SCRT is the preferred RT as part of the total neoadjuvant treatment in Sweden, which is the reason we use it in the ongoing PRORECT trial.

We added line 83 in the Introduction of the manuscript to clarify this.

Comments 2: It is suggested to explicitly state the purpose of this study in last paragraph of Introduction.

Response 2: The purpose of the study is indeed stated in the last paragraph of Introduction. The second sentence starts with ”The purpose of this manuscript…”. This can be found on line 89 of the revised manuscript. No changes made to the manuscript.

Comments 3: Given substantial heterogeneity in LARC (e.g., tumor location, cT/N stage, tumor volume), the sample size (n = 10) of this study raises concerns about generalizability. Could the sample be expanded?

Response 3: Even though we agree with the concerns regarding generalizability, we believe our cohort is representative for LARC. In our cohort 30% of tumors are T4b, 60% are N2, and both tumor height and volume vary greatly between the patients.

Additionally, it is important to point out that the 5×5 Gy, 5×6 Gy, and 5×7 Gy protocol published by Kim et al. (Nonoperative Radiation Management of Adenocarcinoma of the Lower Rectum (NORMAL-R) a prospective, nonrandomized trial ) was based on a cohort of only 20 patients. This dose-escalated protocol has been proven safe and is already in clinical use at several institutions. The main aim of our manuscript is to demonstrate the feasibility of a dose-escalated protocol using protons relative to photons. Therefore, we believe 10 patients is sufficient for our stated purposes. We added lines 305-10 to the revised manuscript at the end of the Discussion in light of this.

Comments 4: Results are currently primarily observational. It would be advisable to incorporate statistical tests to compare the differences among the groups and determine whether they are statistically significant, thereby enhancing the persuasiveness of the findings.

Response 4: We agree with the reviewer that the inclusion of statistical tests would strengthen the manuscript. Statistical tests added to the manuscript (Table 3, lines 222-227, lines 231-233, lines 235-237, Figure 7). 

Comments 5: It is suggested that a section discussing the limitations of this study be added at the end of Discussion.

Response 5: Some limitations are already mentioned in the Discussion section (i.e. that we did not perform any dose accumulations based on daily imaging). However, we agree with the reviewer that this could be expanded. We therefore added a further paragraph at the end of the Discussions section (Lines 305-312). 

Comments 6: In Figure 3, for bowel bag D5cm³, the Protons, PRORECT II group exhibits higher EQD2 doses than other plans, with its upper quartile exceeding tolerance. Please explain this finding and its potential implication.

Response 6: We thank the reviewer for this observation. Lines 197-201 in the Results updated to explain this and suggest a possible clinical implication. Figure 4 appears truncated in the provided PDF. Ensure all labels/legends are fully visible.

Comments 7: Figure 4 appears truncated in the provided PDF. Ensure all labels/legends are fully visible.

Response 7: A new Figure 4 with larger label font size, thicker lines and more clear markers is now in the revised manuscript.

Comments 8: Please check for some typographical errors, such as those on lines 108-109 and line 190, as well as the Figure 7 in line 214.

Response 8: Typographical errors on those lines corrected by Cancers editor. Figure 7 now includes p-values, but otherwise unchanged.

Reviewer 4 Report

Comments and Suggestions for Authors

This study investigates the feasibility of dose-escalated hypofractionated proton therapy for locally advanced rectal cancer, showing that hypofractionated proton therapy is able to achieve target coverage while reducing integral dose and mean OAR dose compared with photon therapy. This planning study successfully demonstrated the feasibility of dose-escalated hypofractionated proton therapy for locally advanced rectal cancer and provided a strong theoretical basis for further clinical research. However, there are still some limitations in this article that need to be improved.

  1. Small sample size. This study includes only 10 patients, limiting statistical power. Larger cohorts are needed to verify the reproducibility, particularly for OAR dose constraints (3/10 photon therapy plans and 2/10 proton therapy plans exceeded OAR limits).

  1. Although dosimetric feasibility has been demonstrated, clinical outcomes (e.g., cCR rates, toxicity) are not evaluated. The association between dose escalation and cCR improvement remains theoretical and requires further trials.

  1. While the results indicate that "two patients" exceeded bowel bag D5cm³ objective and "one patient" exceeded bladder mean dose objective, it does not address whether these exceedances are clinically significant (e.g., risk of toxicity).

  1. The title of the X-axis in Figure 4 is missing.
  2. The median values in Figure 5 are difficult to see.
  3. Line 108,190,242 "Error! Reference source not found" ?

Author Response

Comments 1:  Small sample size. This study includes only 10 patients, limiting statistical power. Larger cohorts are needed to verify the reproducibility, particularly for OAR dose constraints (3/10 photon therapy plans and 2/10 proton therapy plans exceeded OAR limits).

Response 1: While we acknowledge the concerns regarding generalizability, we believe our cohort is representative of patients with locally advanced rectal cancer (LARC).

It is important to note that the dose-escalation protocols of 5×5 Gy, 5×6 Gy, and 5×7 Gy reported by Kim et al. in the NON-OPerative Radiation Management of Adenocarcinoma of the Lower Rectum (NORMAL-R) trial—a prospective, nonrandomized study—were based on a cohort of only 20 patients. This protocol have demonstrated safety and is currently implemented in clinical practice at multiple institutions. While the clinical safety of this dose-escalated fractionation is no longer a clinical concern, the main objective of our manuscript is to evaluate the feasibility of a proton-based dose-escalation strategy in comparison to photon therapy. We believe our data clearly demonstrate that PT can reduce the ID by a factor of two compared to xRT. Furthermore, OAR doses in the PT plans were in general lower than in the xRT plans.

We added lines 305-310 concering this at the end of the Discussion.

Comments 2: Although dosimetric feasibility has been demonstrated, clinical outcomes (e.g., cCR rates, toxicity) are not evaluated. The association between dose escalation and cCR improvement remains theoretical and requires further trials. 

Response 2: We totally agree with this comment. This manuscript is an in-silico preparatory step for the coming randomized clinical trial clinical PRORECT II that will explore the clinical benefits of dose-escalated PT. Currently, we have not treated any patients with dose escalated RT, and therefore we have no clinical outcome data. We added lines 310-312 to this effect at the end of the Discussion.

Comments 3: While the results indicate that "two patients" exceeded bowel bag D5cm³ objective and "one patient" exceeded bladder mean dose objective, it does not address whether these exceedances are clinically significant (e.g., risk of toxicity)

Response 3: We agree with this observation by the reviewer. Lines 197-210 now adress these issues in the manuscript. 

Comments 4: The title of the X-axis in Figure 4 is missing.

Response 4: A new figure 4 generated with larger label font size, thicker lines and more clear markers.

Comments 5: The median values in Figure 5 are difficult to see.

Response 5: A new figure 5 generated with larger label font size, thicker lines and more clear markers. Figure 6 also updated in a similar manner. In addition, it now also uses data that has not be rounded to two decimals. This allows for the median not to overlap with the upper/lower quartiles. As a consequence, lines 230-31 with percentages of voxel having RBE greater than or equal to 1.1 have been updated.

Comments 6: Line 108,190,242 "Error! Reference source not found" ?

Response 6: Corrected by the Cancers editor.

Round 2

Reviewer 3 Report

Comments and Suggestions for Authors

All my concerns have been properly addressed.